# Bioactive Potential of Two Marine Picocyanobacteria Belonging to *Cyanobium* and *Synechococcus* Genera

**DOI:** 10.3390/microorganisms9102048

**Published:** 2021-09-28

**Authors:** Patrizia Pagliara, Giuseppe Egidio De Benedetto, Matteo Francavilla, Amilcare Barca, Carmela Caroppo

**Affiliations:** 1Department of Biological and Environmental Sciences and Technologies, University of Salento, Via Provin-Ciale Lecce-Monteroni, 73100 Lecce, Italy; amilcare.barca@unisalento.it; 2Laboratory of Analytical and Isotopic Mass Spectrometry, Department of Cultural Heritage, University of Salento, 73100 Lecce, Italy; giuseppe.debenedetto@unisalento.it; 3National Research Council, Institute of Heritage Sciences (CNR-ISPC), 73100 Lecce, Italy; 4STAR*Facility Centre, Department of Agriculture, Foods, Natural Resources and Engineering, University of Foggia, 71122 Foggia, Italy; matteo.francavilla@unifg.it; 5National Research Council, Water Research Institute (CNR-IRSA), 74123 Taranto, Italy

**Keywords:** picocyanobacteria toxicity, *Cyanobium*, *Synechococcus*, *Artemia salina*, sea urchin embryos, HeLa cell line, microcystin, BMAA

## Abstract

Coccoid cyanobacteria produce a great variety of secondary metabolites, which may have useful properties, such as antibacterial, antiviral, anticoagulant or anticancer activities. These cyanobacterial metabolites have high ecological significance, and they could be considered responsible for the widespread occurrence of these microorganisms. Considering the great benefit derived from the identification of competent cyanobacteria for the extraction of bioactive compounds, two strains of picocyanobacteria (coccoid cyanobacteria < 3 µm) (*Cyanobium* sp. ITAC108 and *Synechococcus* sp. ITAC107) isolated from the Mediterranean sponge *Petrosia ficiformis* were analyzed. The biological effects of organic and aqueous extracts from these picocyanobacteria toward the nauplii of *Artemia* *salina*, sea urchin embryos and human cancer lines (HeLa cells) were evaluated. Methanolic and aqueous extracts from the two strains strongly inhibited larval development; on the contrary, in ethyl acetate and hexane extracts, the percentage of anomalous embryos was low. Moreover, all the extracts of the two strains inhibited HeLa cell proliferation, but methanol extracts exerted the highest activity. Gas chromatography–mass spectrometry analysis evidenced for the first time the presence of β-*N*-methylamino-l-alanine and microcystin in these picocyanobacteria. The strong cytotoxic activity observed for aqueous and methanolic extracts of these two cyanobacteria laid the foundation for the production of bioactive compounds of pharmacological interest.

## 1. Introduction

Picocyanobacteria (PCCs) (0.2–3 µm) [1] are a conspicuous component of the phytoplankton communities. Unicellular coccoid forms [2,3,4] and prochlorophytes [5,6] represent them. In marine environments, the most represented genera are *Synechococcus*, *Synechocystis*, *Cyanobium*, *Cyanobacterium* [7] and *Prochlorococcus* [8]. PCCs can be detected both in the free-living population in the water column and associated with other organisms (foraminifers, corals, sponges, mollusks) or substrates [9].

PCCs may contribute significantly (i.e., up to 50%) to phytoplankton productivity and biomass in marine waters [10] and they can be responsible for up to 98% of the total biomass in brackish systems [11,12]. PCCs play an important role in the functioning of the microbial loop [13,14], and they actively participate in the modulation of energy and matter flows as well as in the sustenance and development of higher trophic levels [15,16].

However, interest in these microorganisms has recently increased due to the property they have of producing a great variety of secondary metabolites, mainly isolated until now from the filamentous species [17,18,19]. Indeed, studies on PCC biomolecules have developed in more recent years [20,21,22,23,24]. The identified secondary metabolites may have useful properties, such as antibacterial, antiviral, anticoagulant or antitumor activities [25,26,27,28]. These cyanobacterial metabolites have high ecological significance and could be considered responsible for the widespread occurrence of these microorganisms [29]. Their ability to produce bioactive molecules could be associated with ultraviolet radiation protection, anti-predatory defense, allelopathy, resource competition and signaling [22,30]. As an example of anti-predatory defense, *Synechococcus* blooms are known to inhibit zooplankton grazing by producing extracellular polysaccharides [31]. Moreover, *Synechococcus* and *Synechocystis* marine strains are responsible for the production of the microcystin [32,33,34], a toxin considered highly toxic to potential grazers of cyanobacteria [35] and previously detected only in freshwater strains.

The secondary metabolites produced by PCCs comprise oligopeptides, cyanobactines, 2-methylisoborneol (MIB) and geosmin (1,2,7,7-tetramethyl-2-norborneol) (GSM) [20,22,24], some of which are cyanotoxins. The most frequently detected cyanotoxins are hepatotoxins (microcystins, nodularins), neurotoxins (β-methylamino-l-alanine, β-*N*-methylamino-l-alanine, i.e., BMAA) [36,37] and dermotoxins (lipopolysaccharide, LPS) [38].

To date, a limited number of studies explored these small forms of cyanobacteria to identify new bioactive compounds [24,27] also because their isolation is time consuming and costly [20]. However, some PPC species isolated from water samples and substrate collected on beaches were tested to evaluate their ability to produce secondary metabolites with promising applications in cosmetics [39] and pharmaceutics [22,27,40]. Despite the enormous range of biochemicals potentially available from numerous species, currently commercial production of cyanobacteria relies on few genera/species as *Arthrospira, Spirulina*, *Nostoc* and *Aphanizomenon flos-aquae*, primarily used as “health food” or added while manufacturing food supplements and food additives [41,42]. Furthermore, a potential commercial development of cyanobacterial compounds for non-biomedical applications, as herbicides, algicides and insecticides, must be considered [43].

Among the various investigations, the assessment of the cyanotoxin potential of marine PCCs living in association with sponges is particularly interesting, as these invertebrates are considered a huge source of bioactive molecules [44].

Previous investigations on *Cyanobium* and *Synechococcus* strains isolated from the marine sponge *P. ficiformis* demonstrated that these two strains are toxic to *Artemia salina* nauplii and can interfere with sea urchin (*Paracentrotus lividus*) embryonic development [28,45]. Furthermore, toxic effects of these PCC strains have also been evidenced on mussel hemocytes, in which chromatin condensation and fragmentation, typical signs of apoptosis, occurred after cyanobacteria challenge [46].

With the aim to investigate more in depth the bioactivity and chemical nature of secondary metabolites from *Cyanobium* sp. ITAC108 and *Synechococcus* sp. ITAC107, fractionated extracts with different solvent polarity were tested on *A. salina*, sea urchin embryos and human cancer cell lines. Moreover, chemical analyses (GC-MS and LC-MS) were performed to try to identify the compounds responsible for the extract’s bioactivity.

## 2. Materials and Methods

### 2.1. Cyanobacterial Biomass: Culture Conditions, Growth Curve and Harvest

Two cyanobacterial strains, *Synechococcus* sp. ITAC107 and *Cyanobium* sp. ITAC108, previously isolated from the Mediterranean marine sponge *Petrosia ficiformis* [45] were investigated in this study as target species for biological activity screening. Cyanobacterial strain identification was based on both morphological [7,47,48] and molecular criteria as previously described [49].

Cyanobacterial cultures were grown in MN medium enriched with B12 vitamin (5 µg/L) [50]. Culture media was prepared with natural seawater filtered through glass fiber filters (Whatman GF/C, GE Healthcare Company, Maidstone, England) and autoclaved. The cultures were performed in 6 L flasks with 4 L of medium with constant aeration. They were incubated at 24.0 ± 1.0 °C under white fluorescent light at a photosynthetic photon flux of 20 µmol photon^−2^ s^−1^ [51] and an illumination cycle of 18D:6N. 

Picocyanobacteria cells were counted under an epifluorescence microscope (AXIOSKOP ZEISS, Oberkochen, Germany). According to the Guillard and Sieracki [52] method, OD was measured spectrophotometrically at 750 nm with a Varian Cary UV–Visible Spectrophotometer. These data were used to fit a linear regression model between the variables N and OD. The linear correlation between N and OD for *Synechococcus* was y = 8E + 07x − 21,943 (*R*² = 0.99) and for *Cyanobium*, y = 4E + 06x − 35,670 (R² = 0.99), where *y* = N (mL^−1^) and *x* = OD.

Cyanobacteria cells were harvested after seven days of growth (exponential phase) by centrifugation (5100 rpm for 20 min at 5 °C). Concentrated biomass was washed with distilled water to remove NaCl. The solid residue was separated from the supernatant, freeze-dried and stored at −20 °C under N_2_ atmosphere in amber vials until the extraction process.

### 2.2. Fractionated Extraction of Cyanobacterial Biomass

A fractionated extraction was performed on cyanobacterial freeze-dried samples. The extraction procedure was performed according to Francavilla et al. [53]. The idea was to sequentially extract organic compounds from cyanobacterial biomass playing on their difference in polarity. Therefore, four solvents with different polarity indexes (*P*) were used to perform the extractions [54]. n-Hexane (*P* = 0.1), ethyl acetate (*P* = 4.4), methanol (*P* = 5.1) and water (*P* = 10.2) were selected as solvents. Briefly, 0.5 g of ground freeze-dried cyanobacterial biomass was homogenized at 17,000 rpm, using an UltraTurrax T18 IKA homogenizer (IKA®-WERKE GMBH & CO. KG, Staufen, Germany), twice each time with 5 mL of n-hexane at room temperature for 1 min followed by centrifugation. Further extractions in the similar manner were performed sequentially on the pellet using ethyl acetate, methanol and then deionized water (at 80 °C). The isolated supernatants for each solvent (hexane, ethyl acetate, methanol and aqueous) were combined and evaporated under vacuum. The dried extracts were weighed and stored at −20 °C until analysis. To evaluate the effect of the extracts on *A. salina* vitality and on sea urchin development, the dried extracts were dissolved in filtered (0.22 µm) sea water (FSW) to obtain a stock solution of 100 µg/mL.

### 2.3. Acute Toxicity Assay Using Nauplii of Artemia salina

Dried cysts of *A. salina* (JBL NovoTemia, Germany) were hatched in FSW (1 g cysts/L) at 25 °C under continuous illumination and aeration to obtain the nauplii. The *A. salina* nauplii were collected and transferred to 96-well microplates after 24 h of incubation. Ten individuals were transferred to each microplate well containing 100 µL of total volume. A two-fold serial dilution of extracts over 10 wells was performed. Cyanobacteria extracts, prepared as described in the previous paragraphs, were used starting with a dilution of stock solution (50 µg/mL). After 24 h of exposure at 25 °C in darkness, the number of dead larvae in each well was counted. Filtered seawater, methanol, hexane and ethyl acetate were used as negative control. Results are presented as percentage of mortality ± standard deviation (SD) and LC_50_ values was estimated using the Probits statistical method [55].

### 2.4. Sea Urchins and Embryo Toxicity Assay

Adult *P. lividus* sea urchins were collected during the breeding season by SCUBA diving in the Ionian Sea (E 18°00′47″, N 39°58′14″). The sea urchins were acclimated at least for 24 h in natural FSW at 18.0 ± 1 °C (salinity 38.0 ± 0.2‰, pH 8.0 ± 0.2). The field studies did not involve endangered or protected species. All animal procedures were in compliance with the guidelines of the European Union (directive 2010/63/U.E.).

*P. lividus* embryos were obtained as previously described [56]. Briefly, sperm solution was added to an egg suspension, and in vitro fertilization was assured by the observation of the vitelline membrane under a light microscope.

Twenty microliters of the cyanobacterial extracts were added to 20 mL of embryo culture immediately after the vitelline membrane elevation (within 5 min). After 24 and 48 h, aliquots of embryos suspension were fixed with cold methanol and observed under a light microscope (Nikon Eclipse 50i, Tokyo, Japan). One hundred embryos from each treatment were counted to obtain the percentage of normal embryos. Pictures were taken using a camera connected to the microscope.

Three replicates for each extract were performed. The results of the sea urchin embryo toxicity test are reported as the mean of the percentage of deformed embryos ± SD.

The plutei dimensions were evaluated by using the ImageJ program, and differences were obtained by the Student’s *t*-test. Differences were indicated as statistically significant with *P*-values < 0.01.

### 2.5. Cytotoxicity

#### 2.5.1. HeLa Cell Culture Conditions and Treatments

HeLa cells (human cervix adenocarcinoma; cell line ATCC^®^ CCL 2™) were cultured in sterile conditions in 25 or 75 cm^2^ plastic flasks with DMEM, supplemented with 10% (*v*/*v*) FBS, 2 mM l-glutamine and 100 μg/mL penicillin/streptomycin and maintained in 5 % CO_2_ at 37 °C in an incubator (Thermo Fisher Scientific, Waltham, MA, USA). Cells were detached and harvested with a 0.3% (*v*/*v*) trypsin solution and then transferred to new flasks every 2–3 days (70–90% confluence) for propagation. The culture medium was changed every 2 days. All experiments were performed between passage 3 and 10 of propagation. Concentrated extracts were dissolved in DMSO, and the experiments were carried out incubating HeLa cells with DMEM containing 10 µg/mL of different solvent-free extracts. The final concentrations of DMSO did not exceed 0.05%. Cells incubated with fresh medium were used as control.

#### 2.5.2. MTT Test

MTT assay was performed using different extracts (ethyl acetate, methanol and aqueous) from the two cyanobacteria on the HeLa cells. 

Cells were seeded in 96-well plates (20 × 10^3^ cells/well) and incubated for 24 h at 37 °C. After incubation, the medium was removed and replaced with a medium containing aqueous, methanol or acetate extracts (diluted 1:100 in the medium). After treatment, MTT solution (5 mg/mL in sterile filtered PBS, pH 7.4) was added to each well to reach a final concentration of 0.5 mg MTT/mL, and plates were incubated at 37 °C for 3 h. The dark-blue formazan crystals were then solubilized by cell lysis with 200 μL/well 2-propanol/HCl 4N, and absorbance was measured at 550 nm with a Multiskan Fc Microplate Photometer (Thermo Fisher Scientific, Waltham, MA, USA). Data were reported as percentage of control (mean ± SEM) of eight sample replicates per treatment. Three independent experiments were performed.

### 2.6. Secondary Metabolite Determination 

Secondary metabolite analysis was performed by using a gas-chromatograph coupled with a single quadrupole mass spectrometer (GC-MS) apparatus (6890 GC coupled to a 5973 inert MSD, Agilent Technologies, Santa Clara, CA, USA) as already described [57]. Methanolic and aqueous extracts were first methoximated with 10 μL of a 20 mg/mL MeOX solution in pyridine at 40 °C for 90 min and then silylated with 90 μL of MSTFA at 70 °C for 60 min. After cooling at room temperature for 5 min, 1 μL of the solution was injected into the GC-MS, which operated in splitless mode. 

The injector temperature was set at 280 °C. Metabolites were separated on a DB-1ht capillary column 30 m × i.d. 250 μm × 0.1 μm using a continuous flow rate of 1 mL/min of ultrapure helium. The column oven temperature was set at 50 °C for 2 min; then the temperature was increased from 50 to 350 °C with a ramp of 10 °C/min and a hold time of 10 min. The total run time was 42 min. The mass detector was operated at 70 eV in the electron impact (EI) ionization mode. The ion source and transfer line temperatures were set at 250 and 300 °C, respectively, whereas nominal mass scan spectra were acquired with a mass scan range of 50–550 *m*/*z*. 

Fractionated extracts were also analyzed by gas chromatography coupled to high-resolution and high-accuracy mass spectrometry (Thermo Scientific™ Q Exactive™ GC Orbitrap™ GC-MS/MS system, GC-HRMS, Waltham, MA USA) using the same column and gas-chromatograph conditions. Ion source and transfer line temperatures were set at 330 and 280 °C, respectively, HRMS spectra were acquired with a mass scan range of 50–750 *m*/*z*, resolution (FWHM at *m*/*z* 200) was 60,000, mass accuracy 1 ppm using internal lock mass correction and 207.03235 *m*/*z* as lock mass. Low-resolution data were analyzed by Agilent ChemStation, whereas for HRMS measurements, deconvolution, feature identification and putative metabolite identification were carried out using Thermo TraceFinder 4.1 software. Peak deconvolution was automatically performed with TraceFinder, where NIST 2014 and GC-Orbitrap Metabolomics mass spectral libraries were used to annotate the peaks with a search index threshold of >700. Compound identification was made using a total confidence score that considers the NIST spectral match as well as the percentage of fragment ions that can be explained from the elemental composition of the molecular ion assigned by NIST.

Extracts were also analyzed by liquid chromatography coupled with mass spectrometry (LC-MS). A Surveyor MS Pump coupled with an LCQ DECA XP Plus (Thermo Finnigan, Thermo Fisher Scientific, Waltham, MA, USA) ion trap mass spectrometer, equipped with an ESI source, was used for our purposes [58]. A reverse-phase Thermo Scientific Biobasic-C18 column (100 mm × 2.1 mm i.d., particle size 5 µm) with a C18-Security Guard cartridge precolumn (10 mm × 2.1 mm i.d., particle size 5 μm) was used for chromatographic separation at room temperature (25 °C). The injection volume was set at 2 μL. A gradient binary elution was performed using solvent A (0.1% formic acid in water) and solvent B (0.1% formic acid in acetonitrile) at a constant flow of 0.2 mL/min.

The gradient program was set as 5% B (0–5 min), 5–95% B (5–40 min), 95% B (40–45 min), 5% B (45–50 min), 5% B (50–60 min). The mass spectrometer was run in the positive ion mode, and the capillary voltage was set at 3.5 kV. The ion trap scanned the 200–1400 *m*/*z* range during the separation and detection. Automatic gain control was employed using three microscans and a maximum injection time of 140 ms. Electrospray operating conditions were as follows: heated capillary temperature, 280 °C; sheath gas, 55 L min^−1^; auxiliary gas 28 L min^−1^. The spectrometer was calibrated externally with a mixture of caffeine, MRFA and Ultramark according to the manufacturer’s instruction.

Data were acquired using Thermo Xcalibur software, whereas untargeted analysis was carried out using MS-DIAL [59].

## 3. Results

### 3.1. Fractionated Extraction Yields

The yield of fractionated extractions using solvents with different polarity index is showed in Figure 1. The increase in polarity from hexane to methanol was associated with an increase in the extraction yield. Therefore, the highest yields were found using methanol, but different values were observed between the strain *Synechococcus* sp. ITAC107 (87.6% d.w.) and *Cyanobium* sp. ITAC108 (54.4% d.w.). Interestingly, a further increase in polarity achieved by switching from methanol (*P* = 5.1) to water at 80 °C (*P* = 10.2) did not correspond to an increase in extraction yield. Moreover, the water extract was higher for *Cyanobium* sp. ITAC108 (44.1% d.w.) compared to *Synechococcus* sp. ITAC107 (10.6% d.w.). Hexane and ethyl acetate fractions showed lower extraction yields that ranged between 0.02% d.w. (*Cyanobium* sp. ITAC108) and 0.04% d.w. (*Synechococcus* sp. ITAC107) in hexane and 0.08% d.w. (*Cyanobium* sp. ITAC108) and 0.06% d.w. (*Synechococcus* sp. ITAC107) in ethyl acetate, respectively.

### 3.2. Toxicity Test

#### 3.2.1. *Artemia salina*

The toxicity test showed that the extracts from *Synechococcus* sp. ITAC107 and *Cyanobium* sp. ITAC108 were toxic to *Artemia* nauplii after 24 h in a dose-dependent manner. Most of the extracts were found to be toxic, inducing the maximum percentage (100%) of death at the maximum administered dose. As an exception, the ethyl acetate extract of *Cyanobium* sp. ITAC108 did not affect brine shrimp vitality. On the contrary, the most toxic sample was the water extract from *Cyanobium* sp. ITAC108 since its LC_50_ was 3.13 mg/mL (Table 1). For *Synechococcus* sp. ITAC107, we observed that the methanolic extract exerted a stronger activity (6.25 µg/mL) than polar and water fractions. In Table 1, data about the LC_50_ of all the fractions are reported.

#### 3.2.2. Effects on Sea Urchin Embryo Development

Twenty-four hours after fertilization, the control sample presented a prismatic shape. All the cyanobacterial extracts affected the development of sea urchin larvae. Toxicity assays with water and methanolic extracts from *Cyanobium* sp. ITAC108 and *Synechococcus* sp. ITAC107 displayed an evident toxic effect by completely inhibiting embryogenesis (Figure 2): after 24 h, in a sample treated with water and methanolic extracts, most embryos showed, respect to control (Figure 3A), an anomalous development, blocked at the morula stage (Figure 3B,C,E,F). Morulae exhibited anomalous cell division in most cases. On the contrary, in ethyl acetate (Figure 3D) and hexane extracts the percentage of anomalous embryos was low: 21.8 ± 10.6 and 19.4 ± 3.2 for *Cyanobium* sp. ITAC108 and 21.1 ± 15.1 and 32.8 ± 19.5 for *Synechococcus* sp. ITAC107. There were no significant differences (*P* > 0.01) with control, which showed that only 7.3% ± 1.3% were anomalous embryos.

Forty-eight hours after treatment, in control 97.3% ± 1.6 % of sea urchin embryos developed into normal pluteus larvae (Figure 4A,E) with an average length of 518 ± 35 μm. No effects were observed for embryos incubated with hexane and ethyl acetate extracts from *Cyanobium* sp. ITAC108 (Figure 4B). Hexane extract from *Synechococcus* sp. ITAC107 exerted a bland effect on sea urchin embryos that reached the pluteus stage (Figure 4C), but in this case the larval dimensions had a significantly (*P* < 0.01) reduced size (Figure 4F) compared to controls (Figure 4A) with an average length of 338 ± 33 μm. In addition, skeletal deformations (Figure 4F) were recorded in this sample treatment. On the contrary, a very dangerous effect was detected with inhibition of embryonic development after incubation with ethyl acetate extract from *Synechococcus* sp. ITAC107. In the last case, embryos did not survive more than 24 h (Figure 4D), presenting in most cases a completely deformed skeleton (Figure 4G).

#### 3.2.3. Effects on HeLa Cells

In this study, we have explored the effect of various extracts of two cyanobacterial strains on human epithelial-like HeLa cells.

The cytotoxicity of hexane, ethyl acetate, methanol and water extracts from *Cyanobium* sp. ITAC108 and *Synechococcus* sp. ITAC107 was assessed using the MTT assay. As shown in Figure 5, all the extracts inhibited HeLa cell proliferation, but methanol extracts from both cyanobacteria strains exerted the highest activity with 52% and 53% of cell growth inhibition, respectively. Aqueous extracts from the two cyanobacteria also inhibited HeLa growth with a percentage of 48% and 36%, respectively. Cells treated with ethyl acetate extracts were metabolically more active, showing 36% and 32% of inhibition, while for hexane extracts from strain *Cyanobium* sp. ITAC108 and *Synechococcus* sp. ITAC107, 24% and 35% of cell growth inhibition were observed, respectively. 

### 3.3. Secondary Metabolite Determination

Both the methanolic and water extracts from *Cyanobium* sp. ITAC108 and *Synechococcus* sp. ITAC107 were analyzed by GC-MS to identify those metabolites responsible for the toxicity of extracts. The β-*N*-methylamino-l-alanine, BMAA, was identified in the chromatograms of the methanolic extract at 16.71 min as tms derivates using the accurate mass 116.0889 and 291.1263. The relevant extracted ion chromatograms of all the samples are shown in Figure 6, where it is possible to observe, along with BMAA, the presence of 2,4-diaminobutyric acid (2,4-DAB), which also has neurotoxic properties [60].

Appendix A shows all the other annotated metabolites identified in the extracts using GC-HRMS.

The microcystin-VF was identified in LC-ESI(+)-MS at *m*/*z* 972.51 ([M + H]+) in the methanolic extract of both PCCs. As it was not possible to identify other secondary metabolites among those already identified in PCCs [61], an untargeted approach was used to analyze LC-ESI(+)-MS data. As methanolic extracts showed a greater activity toward HeLa cells than aqueous extracts did, during the untargeted analysis carried out using MS-DIAL, methanolic and aqueous extracts were considered as different classes. Appendix A reports the PCA score plot of the samples, where it is possible to see that methanolic extracts are separated from aqueous along PC1, whereas PC2 permits the separation of *Cyanobium* sp. ITAC108 from *Synechococcus* sp. ITAC107 aqueous extracts.

Appendix A lists all the significant (*P*-values < 0.05) characteristics that were identified and putatively annotated. Data curation is beyond the scope of the present paper; however, it is evident that there are different classes of compounds characterizing the extracts and possibly having a role in the recorded PCC activity.

## 4. Discussion

In the present investigation, we focused our attention on two marine strains of PCCs: *Cyanobium* sp. ITAC108 and *Synechococcus* sp. ITAC107, already characterized by morphological and molecular criteria [49]. Previous investigation on these two PCC strains, suggested better characterization of their potential as bioproducers given their ability to affect HeLa cell vitality [49] and to gain more insight into the nature of the more active molecules. The potential as producers of bioactive compounds of some strains of *Synechococcus* and *Cyanobium* genera was previously reported by some authors [27,62,63,64] who emphasize the importance of also studying marine PCCs [22,24,27]. As unicellular coccoid forms, these two PCCs were easily cultured in the laboratory, requiring less care and exhibiting a faster growth rate than the filamentous forms. These properties make them good candidates as bioproducers.

The first step toward the identification of the chemical nature of these compounds was the fractionation of the crude extracts, which suggested that both samples contain a very low amount of apolar (lipophilic) compounds (aldehydes and ketones, hydrocarbons, esters, short chain alcohols, etc.). However, these types of compounds from both cyanobacteria strains were found to be toxic to *A. salina* at the highest concentration used (50 µg/mL) except for the *Cyanobium* sp. ITAC108 ethyl acetate sample. On the other hand, a very high content of polar compounds likely phenols, ammines, amides, carboxylic, etc. (87.6% d.w.) was found in *Synechococcus* sp. ITAC107, while *Cyanobium* sp. ITAC108 contained a significantly lower amount (54.4%) of them. All these fractions were toxic to brine shrimp. Methanolic extracts from the two strains showed comparable activity. 

During the fractionated extraction, hot water was used to solubilize very polar and protic compounds including most likely carbohydrates, polymers and proteins. *Cyanobium* sp. ITAC108 seems to contain a significantly higher amount of this latter class of compounds compared to the sample from *Synechococcus* sp. ITAC107. The amount of protic and polar compounds reflects the stronger toxicity of *Cyanobium* sp. ITAC108 toward *Artemia* nauplii compared to that of *Synechococcus* sp. ITAC107. Although in some cases *Cyanobium* strains did not demonstrate toxicity for *A. salina* [27], other studies [63,65] showed that one strain of *Cyanobium* and one of *Synechococcus* caused acute toxicity in the brine shrimp nauplii, inducing 100% mortality with aqueous and methanolic extracts.

The toxic effect of the different extracts of the cyanobacterial strains was also evident for the embryos of the sea urchin, *P. lividus*. After treatment with water and methanolic extracts from both cyanobacteria strains, embryogenesis stopped at the first stages (segmentation) of development (within 24 h), providing strong evidence for the presence of polar compounds interfering with growth factors. Affecting the segmentation phase of the embryo, we may suppose that both extracts interfere with cell mitotic activity. 

For some extracts (i.e., hexane), embryogenesis of the sea urchin embryos was not inhibited. However, after a longer treatment with hexane extract from *Synechococcus* sp. ITAC107, the resulting larvae were significantly smaller than the control larvae. This suggests that a chronic effect may occur with this hexane extract and that the reduced size of the larvae could be due to the action of compounds present in low concentrations. Sellem and coworkers [66] also suggested this possibility when they tested the effects of different concentrations of unsaturated fatty acids of a dinoflagellate species to embryos of the sea urchin *P. lividus*.

Besides invertebrates, toxicity is often evaluated on mammalian cells by using cancer cell lines [67,68,69]. Interestingly, some study reported a good correlation between the toxic activity evidenced by the *A. salina* test and that observed with tumor cell lines [70]. Here, we report about the effect of the various fractionated extracts on the human cervical cancer HeLa cell line that was already affected by the total crude extracts from both PCCs [49]. The toxicity was checked using the MTT assay, which showed that the cells were metabolically less active in the presence of all the extracts in comparison to control (without extracts). Although the response of cells to various extracts and the percentage change in metabolic activity was different depending on the extract type, methanolic fractions from both PCCs exerted the highest negative effect. The different responses may be due to the selective cellular response and affinity to compounds present in the various extracts.

Methanolic extracts from other cyanobacteria strains are known to induce an antiproliferative effect on HeLa cells [71], but to our knowledge, this is the first time that different fractionated extracts from these two strains were tested on Hela cells. 

All these results highlighting a strong bioactivity of the methanolic and aqueous extracts of both PCCs suggested analyzing them by GC-MS and identifying those metabolites responsible for the toxicity of the extracts. 

All four extracts of the two PCCs had various bioactive components possessing numerous activities. BMAA is one of the important compounds that we found in methanolic extracts along with 2,4-DAB, whereas it was not possible to identify the other BMAA constitutional isomer, i.e., N-(2-aminoethyl)-glycine (AEG). BMAA is a neurotoxin previously identified by other researchers in symbionts and free-living cyanobacteria [68]. *Prochlorococcus marinus*, *Synechoccous* spp. and *Synechocystis* spp. [22,72,73] represent the PCC strains until now identified as BMAA producers

Here, we report for the first time the presence of BMAA and one of its constitutional isomers, 2,4-DAB, in a *Cyanobium* and in a *Synechoccoccus* strain isolated from a marine invertebrate (i.e., the sponge *Petrosia ficiformis*) that could be the main compound responsible for the observed toxicity. 

LC-MS also revealed the presence of the hepatotoxic microcystins (MCs) in our samples. Microcystins, representing the main secondary metabolites here identified, are cyclic heptapeptides typically produced by cyanobacteria that could be the main products influencing the survival of different model organisms we used. These cyanotoxins have been already found in PCCs as reported in Śliwińska-Wilczewska and coauthors [22]. According to their review, the most common MC isomers are MC-LR and MC-YR, while the MC-VF identified in our strains was already detected only in the filamentous *Microcystis aeruginosa* [74]. Other secondary metabolites produced by PCCs are 2-MIB and GSM [20]. However, both compounds were not identified either in the methanolic or in the aqueous extracts.

Finally, considering that the two strains used here have shown high growth rates in laboratory cultures, the metabolites of their active fractions could be considered as easily accessible by-products from biomass that can increase the value of their large-scale culture. However, it is known that the genomic organization of biosynthetic gene clusters, complex gene expression patterns and low yields of compounds synthesized by native producers restrict access to many of these valuable molecules for detailed studies [75]. Recently some investigations have highlighted the importance of genetic engineering in this field. An example of this is represented by the production of key compounds of the hapalindole family of indole-isonitrile alkaloids obtained by engineering the fast-growing cyanobacterium *Synechococcus elongatus* UTEX 2973 [75].

## 5. Conclusions

This work provides further useful data to expand the range of cyanobacteria from which new compounds with significant bioactivity could be identified. Particularly intriguing was the bioactivity of polar extracts (especially the methanolic ones) of two PCC strains belonging to *Cyanobium* and *Synechococcus* genera. To our knowledge, this is the first time in which BMAA, 2,4-DAB and microcystin have been found in cyanobacteria isolated from a marine sponge. We consider these substances to be responsible for cytotoxicity and antimitotic activity, but further investigations are needed to isolate and better characterize these compounds. Furthermore, it will be important to deepen the observations on the long-term effects as the extracts of ethyl acetate, which, although not having antimitotic activity, block the development of the sea urchin by preventing the gastrulation phase. Furthermore, future investigations could help to know the cause of the skeletal deformation induced by the hexane extract from *Synechococcus* sp. ITAC 107. According to our results, we consider the active fractions to be promising for the isolation of compounds for potential biotechnological applications, including the field of pharmacological agents. In addition, considering that the two strains used here have shown high growth rates in laboratory cultures, the metabolites of their active fractions can be considered as easily accessible by-products from biomass that can increase the value of their large-scale culture. However, although a high growth rate has been shown in the laboratory, large-scale production of bioproducts is likely to be best achieved by engaging in the engineering of these cyanobacteria. Another challenge to be faced in the near future. 

## Figures and Tables

**Figure 1 microorganisms-09-02048-f001:**
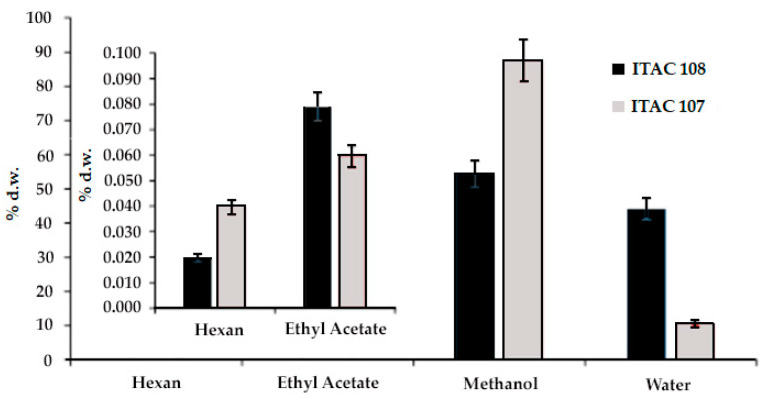
Fractionated extraction yield (expressed as % dry weight) of *Cyanobium* sp. ITAC108 and *Synechococcus* sp. ITAC107 biomass. Different solvents with different polarity indexes (hexane, ethyl acetate, methanol and water) were used.

**Figure 2 microorganisms-09-02048-f002:**
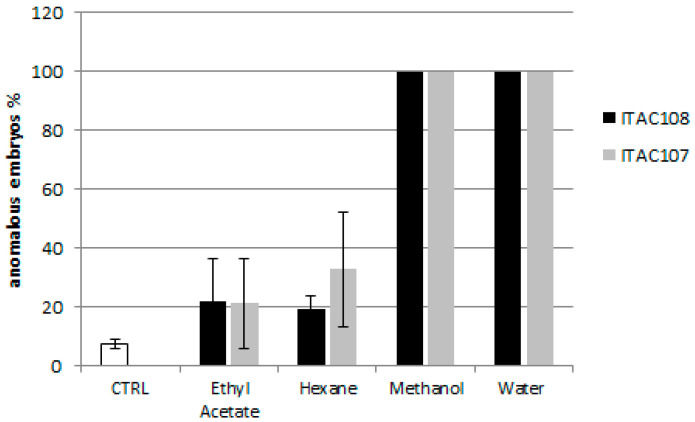
Percentage of anomalous sea urchin embryos after 24 h of treatment with cyanobacterial extracts.

**Figure 3 microorganisms-09-02048-f003:**
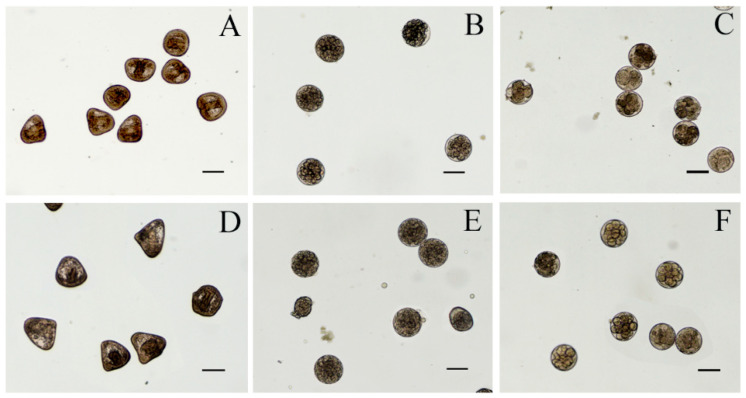
Micrographs of *P. lividus* embryos after 24 h of treatment with cyanobacteria extracts. (**A**) Control sample; (**B**,**C**) sea urchin embryos treated with aqueous extracts from *Cyanobium* sp. ITAC108 and *Synechococcus* sp. ITAC107, respectively; (**D**) sea urchin embryos treated with hexane extracts from *Synechococcus* sp. ITAC107; (**E**,**F**) sea urchin embryos treated with methanolic extracts from *Cyanobium* sp. ITAC108 and *Synechococcus* sp. ITAC107, respectively. Bar represents 100 µm.

**Figure 4 microorganisms-09-02048-f004:**
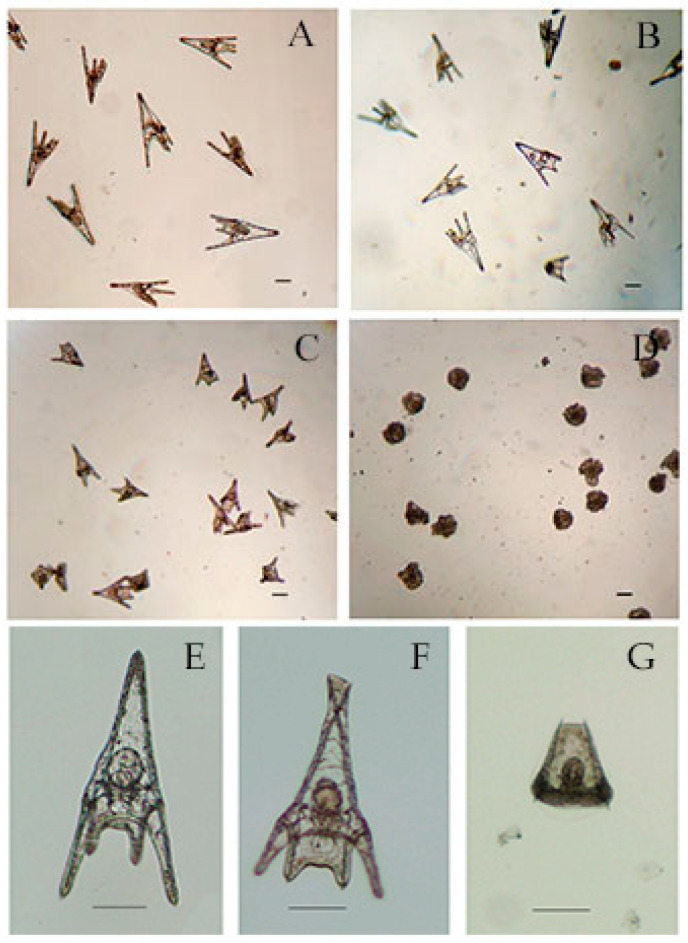
Micrographs of *P. lividus* embryos after 48 h of treatment with cyanobacteria extracts. (**A**) Control sample. (**B**) Sea urchin embryos treated with ethyl acetate extracts from *Cyanobium* sp. ITAC108. (**C**) Sea urchin embryos treated with hexane extracts from *Synechococcus* sp. ITAC107. (**D**) Sea urchin embryos treated with ethyl acetate extracts from *Synechococcus* sp. ITAC107. (**E**) Untreated *P. lividus* embryo. (**F**) Embryo treated with ethyl acetate extracts from *Synechococcus* sp. ITAC107. (**G**) Embryo treated with ethyl acetate extracts from *Synechococcus* sp. ITAC107. Bar represents 100 µm.

**Figure 5 microorganisms-09-02048-f005:**
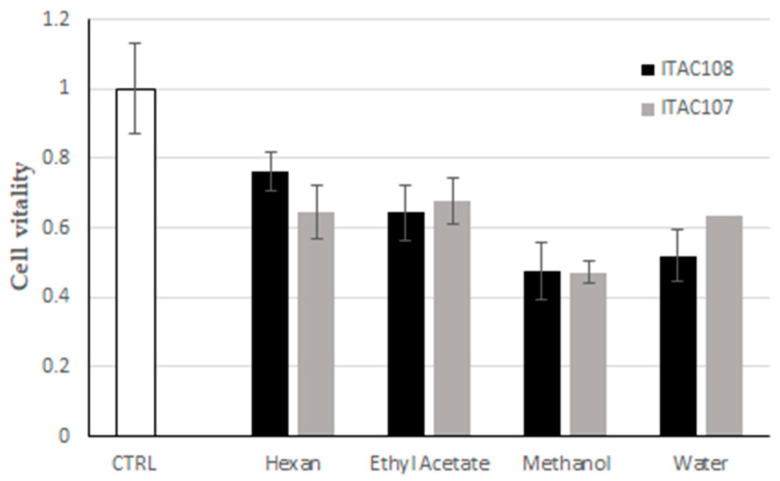
Effect of cyanobacterial fractions on cultured cells viability. The MTT assays was performed on human HeLa cells exposed for 6 h to cyanobacterial fractions from *Synechococcus* sp. ITAC107 and *Cyanobium* sp. ITAC108 strains.

**Figure 6 microorganisms-09-02048-f006:**
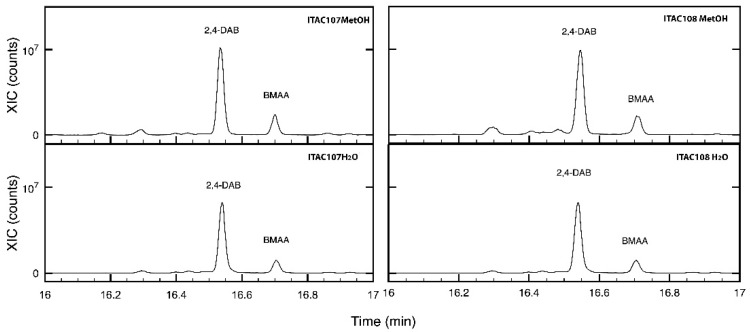
Extracted chromatograms of *m*/*z* 291.1263 ion for methanolic and aqueous extracts for *Cyanobium* sp. ITAC108 and *Synechococcus* sp. ITAC107 in the region comprised between 16 and 17 min. In the chromatograms, it is possible to identify 2,4-DAB and BMAA.

**Table 1 microorganisms-09-02048-t001:** The 24 h LC_50_ values (with 95% confidence limits) for *A. salina* exposed to cyanobacterial extracts.

Strains	LC_50_ 24 h µg/mL	95% Confidence Limits
*Cyanobium* sp. ITAC108 MetOH	6.25	4.71–8.29
*Cyanobium* sp. ITAC108 Water	3.13	1.87–4.76
*Cyanobium* sp. ITAC108 Hexane	9	7–11
*Cyanobium* sp. ITAC108 Ethyl Acetate	-	-
*Synechococcus* sp. ITAC107 MetOH	6.25	3.96–9.86
*Synechococcus* sp. ITAC107 Water	10.19	5.93–17.55
*Synechococcus* sp. ITAC107 Hexane	7	4–11
*Synechococcus* sp. ITAC107 Ethyl Acetate	16.3	13–20.67

## Data Availability

Most of the data presented in this study are contained within the article or Appendix A; data not present are available on request from the corresponding author.

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
