# Peer review of "Bioactive Potential of Two Marine Picocyanobacteria Belonging to Cyanobium and Synechococcus Genera"

_microorganisms, 2021, doi:10.3390/microorganisms9102048_

Round 1
Reviewer 1 Report
Most of the suggested changes have been included in the revised version.
It could be accepted in the current state.
Reviewer 2 Report
Usually, in the MDPI reviewing process, authors should provide point-by-point answers to reviewers
Did not see any file associated with the revised MS....
This manuscript is a resubmission of an earlier submission. The following is a list of the peer review reports and author responses from that submission.
Round 1
Reviewer 1 Report
The authors report on a bioprospecting study focused in the secondary metabolites produced by the cultures of picocyanobacteriaCyanobium sp. ITAC108 and Synechococcus sp. 21 ITAC107 isolated from the Mediterranean sponge Petrosia ficiformis.Thus, the biological effects of organic and aqueous extracts from these picocyanobacteria towards the nauplii of Artemia salina, sea urchin embryos of Paracentrotus lividusand human cancer lines HeLa cells, were evaluated. All the extracts of the two strains inhibited HeLa cell proliferation but with moderate activity. Gas chromatography-mass spectrometry analysis evidenced the presence of β-N-methylamino-L-alanine (BMAA), 2,4-diamonobutyric acid (2,4-DAB) and microcystin in these picocyanobacteria.
1.- Microcystins are highly toxic substances due to their phosphatase inhibition effects. It does not seem rational that the samples show moderate toxicity with mycocystins present. Therefore, more evidence of the presence of microcytines should be provided.
2.- Figure 7 with PCA is irrelevant. Could be removed or included only in supplemental material.
3.- BMAA and 2,4-DAB are readily available as commercial products. One could compare the effect of these pure compounds with that of the extracts to see if they are responsible for the activity.
4.- 2,4-DAB is L or D?
Reviewer 2 Report
Peer-review
Bioactive potential of two marine picocyanobacteria belonging to Cyanobium and Synechococcus genera
Coccoid cyanobacteria produce a great variety of secondary metabolites, which may have useful properties, as antibacterial, antiviral, anticoagulant or anticancer.
What kind of commercial products from cyanobacteria are already available at large industrial scale on the world market?
except those coming from Arthrospira/Spirulina
Cyanobium and Synechococcus
are easy to produce? which kind of reactors?
work conducted on methanolic and aqueous extracts from the two strains
only gas chromatography-mass spectrometry analysis?
what about non-volatile compounds?
2.3. Acute toxicity assay using nauplii of Artemia salina
2.4. Sea urchins and embryo toxicity assay
2.5. Cytotoxicity
2.5.1. HeLa cell culture conditions and treatments
2.5.2. MTT test
this is unsatisfactory to investigate these assays with extracts, fractions
I would recommend bio-guided fractionation up to pure compounds
working only with fractions: one day it works, you produce again your biomass, test again, it doesn’t work
pure compound identified with true biological properties, could be monitored
Therefore, the highest yields were found using metha- 240
nol, but different values were observed between the strain Synechococcus sp. ITAC107 241
(87.6% d.w.) and Cyanobium sp. ITAC108 (54.4% d.w.). Interestingly, a further increase in 242
polarity achieved switching from methanol (P=5.1) to water at 80°C (P=10.2) did not cor- 243
respond to an increase in extraction yield. Moreover, the water extract was higher for 244
Cyanobium sp. ITAC108 (44.1% d.w.), compared to Synechococcus sp. ITAC107 (10.6% 245
d.w.). Hexane and ethyl acetate fractions showed lower extraction yields that ranged be- 246
tween 0.02% d.w. (Cyanobium sp. ITAC108) and 0.04% d.w. (Synechococcus sp. ITAC107) 247
in hexane and 0.08% d.w. (Cyanobium sp. ITAC108) and 0.06% d.w. (Synechococcus sp. 248
ITAC107) in ethyl acetate, respectively.
It is usal that methanol extraction yields are far higher than Hexane and ethyl acetate extraction yields
What is the lipids content of your strains??
As unicellular coccoid forms, 366
these two PCC were easily cultured in the laboratory requiring less care and exhibiting a 367
faster growth rate than the filamentous forms. These properties make them good candi- 368
dates as bioproducers.
unicellular coccoid forms already cultivated at multi m3 volumes?
LCMS also revealed the presence of the hepatotoxic microcystins (MCs) in our sam- 431
ples. Microcystins, representing the main secondary metabolites here identified, are cyclic 432
heptapeptides typically produced by cyanobacteria that could be the main products in- 433
fluencing the survival of different model organisms we used.
LCMS? not presented in M&M?
Authors expect high-enough production level of bioactive metabolites? without genetic engineering?
Engineered Production of Hapalindole Alkaloids in the Cyanobacterium Synechococcus sp. UTEX 2973
Open Access
Knoot, C.J., Khatri, Y., Hohlman, R.M., Sherman, D.H., Pakrasi, H.B. 2019 ACS Synthetic Biology
8(8), pp. 1941-1951
Exploring for a long time…. industrial issues??
Exploring marine cyanobacteria for lead compounds of pharmaceutical importance
Open Access
Uzair, B., Tabassum, S., Rasheed, M., Rehman, S.F. 2012 The Scientific World Journal
2012,179782
ULTIMATE GOAL, a pure, original bioactive compound
Antitumor Activity of Hierridin B, a Cyanobacterial Secondary Metabolite Found in both Filamentous and Unicellular Marine Strains
Open Access
Leão, P.N., Costa, M., Ramos, V., (...), Vasconcelos, V.M., Martins, R. 2013 PLoS ONE
8(7),e69562